# Identifying the Pathogenic Variants in Heart Genes in Vietnamese Sudden Unexplained Death Victims by Next-Generation Sequencing

**DOI:** 10.3390/diagnostics14171876

**Published:** 2024-08-27

**Authors:** Tho Nguyen Tat, Nguyen Thi Kim Lien, Hung Luu Sy, To Ta Van, Duc Dang Viet, Hoa Nguyen Thi, Nguyen Van Tung, Le Tat Thanh, Nguyen Thi Xuan, Nguyen Huy Hoang

**Affiliations:** 1Department of Forensic Medicine, Hanoi Medical University, 1 Ton That Tung Str., Dongda, Hanoi 100000, Vietnam; nguyentattho1979@gmail.com (T.N.T.); drlshung@gmail.com (H.L.S.); 2Institute of Genome Research, Vietnam Academy of Science and Technology, 18-Hoang Quoc Viet Str., Caugiay, Hanoi 100000, Vietnam; ntkimlienibt@gmail.com (N.T.K.L.); tungnv53@gmail.com (N.V.T.); thanhlt@igr.ac.vn (L.T.T.); xtltam76@gmail.com (N.T.X.); 3Department of Pathology, National Cancer Hospital, 43 Quan Su Str., Hoan Kiem, Hanoi 100000, Vietnam; tavanto.bvk@gmail.com; 4Cardiovascular Intensive Care Unit, Heart Institute, 108 Military Central Hospital, 1B Tran Hung Dao Str., Hai Ba Trung, Hanoi 100000, Vietnam; dangvietduc108@gmail.com; 5Faculty of Biotechnology, Graduate University of Science and Technology, Vietnam Academy of Science and Technology, 18 Hoang Quoc Viet Str., Caugiay, Hanoi 100000, Vietnam; nguyenhoatn212@gmail.com

**Keywords:** cardiac channelopathies, cardiomyopathies, molecular autopsy, next-generation sequencing (NGS), sudden unexplained death (SUD), Vietnamese victims

## Abstract

In forensics, one-third of sudden deaths remain unexplained after a forensic autopsy. A majority of these sudden unexplained deaths (SUDs) are considered to be caused by inherited cardiovascular diseases. In this study, we investigated 40 young SUD cases (<40 years), with non-diagnostic structural cardiac abnormalities, using Targeted NGS (next-generation sequencing) for 167 genes previously associated with inherited cardiomyopathies and channelopathies. Fifteen cases identified 17 variants on related genes including the following: *AKAP9*, *CSRP3*, *GSN*, *HTRA1*, *KCNA5*, *LAMA4*, *MYBPC3*, *MYH6*, *MYLK*, *RYR2*, *SCN5A*, *SCN10A*, *SLC4A3*, *TNNI3*, *TNNI3K*, and *TNNT2*. Of these, eight variants were novel, and nine variants were reported in the ClinVar database. Five were determined to be pathogenic and four were not evaluated. The novel and unevaluated variants were predicted by using in silico tools, which revealed that four novel variants (c.5187_5188dup, p.Arg1730llefsTer4 in the *AKAP9* gene; c.1454A>T, p.Lys485Met in the *MYH6* gene; c.2535+1G>A in the *SLC4A3* gene; and c.10498G>T, p.Asp3500Tyr in the *RYR2* gene) were pathogenic and three variants (c.292C>G, p.Arg98Gly in the *TNNI3* gene; c.683C>A, p.Pro228His in the *KCN5A* gene; and c.2275G>A, p.Glu759Lys in the *MYBPC3* gene) still need to be further verified experimentally. The results of our study contributed to the general understanding of the causes of SUDs. They provided a scientific basis for screening the risk of sudden death in family members of victims. They also suggested that the Targeted NGS method may be used to identify the pathogenic variants in SUD victims.

## 1. Introduction

In forensic medicine, an autopsy has determined the cause of only two-thirds of sudden deaths, and the remaining cases are called sudden unexplained death (SUD) [1]. The majority of SUD cases are believed to be caused by cardiovascular diseases leading to sudden cardiac death (SCD) in young people. Sudden death is usually defined as a person who appeared healthy for 24 h before the onset of symptoms that led to death [2]. SCD accounts for 15–20% of all deaths in the general population, with incidence ranging from 40 to 100 per 100,000 people annually [3]. The main causes of SCD include cardiomyopathies, channelopathies, and ischemic heart diseases. About two-thirds of SCD cases had structural abnormalities that were evident at autopsy [1,4,5]; meanwhile, no structural abnormalities of the heart were found in the remaining cases. SCD may result from complications of cardiomyopathy causing early malignant arrhythmias, leading to death before the development of the cardiomyopathy phenotype. Especially in young people, SCD is often due to cardiomyopathies such as hypertrophic cardiomyopathy (HCM), dilated cardiomyopathy (DCM), and arrhythmogenic cardiomyopathy (ACM); channelopathies including ion channel disorders such as long QT syndrome (LQTS), short QT syndrome (SQTS), Brugada syndrome (BrS), catecholaminergic polymorphic ventricular tachycardia (CPVT), progressive cardiac conduction disorder, and early repolarization syndrome; or ischemic heart diseases [6,7,8].

In atrioventricular canal disease, long QT syndrome (LQTS) is caused by a heart’s electrical system that takes too long to recharge. The disease can lead to life-threatening arrhythmias and sudden cardiac death, primarily at a young age [9]. At least 15 genes encoding different ion channels have been identified as causes of LQTS [10]. Short QT syndrome (SQTS) is considered one of the most dangerous diseases associated with sudden cardiac death, characterized by a short QT interval on the electrocardiogram. Potentially pathogenic variants have been reported in five genes (*CACNA2D1*, *KCNH2*, *KCNJ2*, *KCNQ1*, and *SLC4A3*); all are inherited in an autosomal dominant [2,11,12]. Brugada syndrome (BrS) is a cardiac channelopathy caused by rare mutations in the *SCN5A* gene, which encodes the alpha-subunit of the voltage-dependent cardiac Na+ channel protein (Nav1.5) [2,13]. SCN5A-positive BrS patients often exhibit severe conduction abnormalities [14] and have more severe arrhythmic outcomes than SCN5A-negative patients [15,16].

CPVT is a malignant arrhythmia syndrome characterized by bidirectional or polymorphic VT during physical or emotional stress, which leads to a significantly high mortality rate (30% SCD before age 40) if the patient is not detected and treated [17]. Mutations in the *RYR2* gene were found in ≈60% of patients with CPVT [18] and concentrated mainly in three specific regions of the ryanodine receptor 2 (RYR2) [18,19]. Additionally, mutations in *CASQ2* have been identified as causing a less common but more severe form of CPVT [20]. Mutations in *RYR2* and *CASQ2* result in diastolic calcium release from the SR and cause arrhythmias. Recently, two other genes, *CALM1* and *TRDN*, involved in calcium balance, have also been identified as causing CPVT [21,22].

In addition to arrhythmias, progressive cardiac conduction disorders, early repolarization syndrome, and ischemic heart disease are also of concern and are considered causes of sudden death. Progressive cardiac conduction disorder (PCCD), characterized by delayed impulse conduction through the His–Purkinje system with right or left bundle branch block, is a heart disease leading to complete atrioventricular (AV) block, syncope, and SCD. Genetic forms of PCCD often overlap or co-occur with other heart diseases. Currently, 20 genes encoding cardiac ion channels and regulatory proteins, protein kinases, structural proteins, and transcription factors are associated with different forms of PCCD [23]. In recent years, early repolarization syndrome (ERS), which is caused by variants in eight ion channel genes (*KCNJ8*, *ABCC9*, *SCN5A*, *SCN10A*, *KCND3*, *CACNA1C*, *CACNB2b*, and *CACNA2D1*), has been considered a hereditary SCD syndrome [24,25,26,27,28,29]. In addition, cerebral small vessel disease (CSVD), a clinically and genetically heterogeneous group of disorders, is also considered a leading cause of myocardial infarction and sudden death [30]. Although most cases of CSVD are sporadic, recent reports have identified genes associated with CSVD, including *NOTCH3*, *HTRA1*, *CTSA*, *GLA*, *COL4A1/A2*, *TREX1*, and *CSF1R* [31].

Other causes of SCD were related to cardiomyopathy including hypertrophic cardiomyopathy (HCM), dilated cardiomyopathy (DCM), and arrhythmogenic cardiomyopathy (ACM). HCM is the most common form that leads to left ventricular outflow tract obstruction, coronary ischemia due to the narrowing of small blood vessels, and severe arrhythmias [32], which is the main cause of sudden cardiovascular death [33]. Most of these cases are caused by pathogenic variants in the core sarcomeric genes (*MYH7*, *MYBPC3*, *TNNT2*, *TNNI3*, *MYL2*, *MYL3*, *TPM1*, *ACTC1*) [34]. To date, about 100 genes associated with cardiomyopathy have been reported [35].

An autopsy aims to determine the cause of death in the victims and predict the risks to family members. However, until now, in 40% of SUD cases, the causes could not be determined by autopsy [36]. Therefore, genetic testing has been proposed in forensic medicine to investigate sudden death through genetic analysis [37]. It is especially useful in cases of negative traditional autopsies or SUD due to underlying genetic arrhythmic heart disease. Early molecular autopsies relied on Sanger sequencing. Although this is an accurate and easy-to-use method, it is limited because it has low throughput and can only be used to analyze a subset of small genes. Recently, next-generation sequencing (NGS) technologies have allowed exome/genome examination, providing an increase in the detection of pathogenic variants and the discovery of newer genotype–phenotype associations [38,39,40]. NGS can investigate large numbers of genes and thus aid in clinical investigations and increase the likelihood of determining the cause of death.

In this study, we performed Targeted NGS sequencing in autopsy-negative SUD victims to identify genetic alterations associated with cardiomyopathy and channelopathies that may explain their causes of death.

## 2. Materials and Methods

### 2.1. Subjects

A total of 40 young SUD cases (30 men and 10 women) collected from the Department of Forensic Medicine, Hanoi Medical University, Vietnam, were investigated for molecular forensics. The victims were all <40 years old (from 1 to 40 years old, median age of 28.85) and died suddenly. Autopsy evaluations as well as toxicology screening were negative. The victims were known to be healthy 24 h before death, and none had a family history of SCD (Table 1).

### 2.2. Targeted Next-Generation Sequencing

To determine the cause of sudden death in the victims, we performed Targeted next-generation sequencing using a gene panel of 167 genes (Appendix A) associated with cardiomyopathy and channelopathies. Genomic DNA was extracted from blood samples using the Qiagen DNA mini kit (QIAGEN, Hilden, German) following the manufacturer’s instructions. DNA concentration was determined using a Thermo Scientific NanoDrop spectrophotometer (Waltham, MA, USA). Sequencing was performed on a Nextseq 500 (Illumina, CA, USA). Variants were compared and annotated based on the GRCh38/hg19 reference genome sequence. Variants in the genes associated with cardiomyopathy and channelopathies were identified based on ACMG guidelines (The American College of Medical Genetics and Genomics) [41].

### 2.3. Sanger Sequencing

Sanger sequencing was performed to confirm the variants were found in these cases. PCR products were purified with the Qiagen PCR Purification kit (QIAGEN, Hilden, Germany) and sequenced on the ABI PRISM 3500 Genetic Analyzer machine (Thermo Fisher Scientific Inc., Waltham, MA, USA) in both directions using the primers that were used in the initial PCR reaction. The sequencing data were analyzed using BioEdit 7.2.5 software.

### 2.4. In Silico Prediction

The influence of any novel nucleotide changes was evaluated with the following in silico prediction tools: CADD [https://cadd.gs.washington.edu/snv; accessed: 1 July 2024], FATHMM [http://fathmm.biocompute.org.uk/inherited.html; accessed: 1 July 2024], Mutation Taster [https://www.mutationtaster.org/ accessed: 2 July 2024], PhD-SNP [https://snps.biofold.org/phd-snp/phd-snp.html; accessed: 1 July 2024], PolyPhen 2 [http://genetics.bwh.harvard.edu/pph2/; accessed: 1 July 2024], SNP&GO [https://snps-and-go.biocomp.unibo.it/snps-and-go/; accessed: 1 July 2024] for misene variants and EX-SKIP [https://ex-skip.img.cas.cz/; accessed: 1 July 2024], Fruitfly [https://www.fruitfly.org/seq_tools/splice.html; accessed: 1 July 2024], MaxEntScan [http://hollywood.mit.edu/burgelab/maxent/Xmaxentscan_scoreseq.html; accessed: 1 July 2024], NetGene2 v.2.42 [https://services.healthtech.dtu.dk/; accessed: 1 July 2024], and Spliceailookup [https://spliceailookup.broadinstitute.org/; accessed: 1 July 2024] for splicing variants.

## 3. Results

In this study, 40 Vietnamese SUD victims were investigated through molecular forensics by using Targeted NGS sequencing. The detected variants were confirmed by Sanger sequencing in these cases (Figure 1 and Figure 2). The results showed that 25 (62.5%) victims harbored no variants or benign variants, and 15 (37.5%) victims had 17 variants identified on the genes associated with cardiomyopathy and channelopathies. Among them, eight of these variants were novel, five variants were pathogenic variants in the ClinVar database, and four variants were published in ClinVar but with uncertain significance (Table 2). Seven variants (41%) were detected in genes associated with cardiomyopathies including *CSRP3*, *LAMA4*, *MYH6*, *MYBPC3*, *TNNI3*, *TNNI3K*, and *TNNT2*. Eight variants (47%) were detected in genes related to cardiac channelopathies such as *AKAP9*, *GSN*, *KCNA5*, *SCN5A*, *SCN10A*, *SLC4A3*, and *RYR2*. Two other variants were detected in the *HTRA1* and *MYLK* genes that were reported in patients with CSVD and extensive aortic, respectively. The novel variants and variants not evaluated on the ClinVar database were predicted by prediction tools. Prediction results showed that four novel variants including variants c.10498G>T, p.Asp3500Tyr (in the *RYR2* gene); c.5187_5188dup, p.Arg1730llefsTer4 (in the *AKAP9* gene); c.1454A>T, p.Lys485Met (in the *MYH6* gene); and c.2535+1G>A (in the *SLC4A3* gene), and an uncertain significance variant c.292C>G, p.Arg98Gly in the *TNNI3* gene, are considered likely to cause disease (A and B in Table 3 and Appendix A). In addition, variant c.683C>A, p.Pro228His in the *KCNA5* gene was predicted as disease-causing by the following in silico tools: CADD (with score 24.1), Mutation taster (with score 77), Polyphen 2 (with score 1.000), and SNP&GO (with score R17). Two variants c.2275G>A, p.Glu759Lys (in the *MYBPC3* gene) and c.5025A>T, p.Glu1675Asp (in the *LAMA4* gene) were predicted as disease-causing by the following in silico tools: CADD (with score 26.7 and 23.6, respectively), Mutation taster (with score 56 for variant in *MYBP3*), PhD-SNP (with score R14 for variant in *LAMA4*), and Polyphen 2 (with score 1.000 and 0.998, respectively). Variants c.298C>T, p.Arg100Cys (in *CSRP3)*, c.4840G>A, p.Glu1614Lys (in *MYLK)*, and c.872T>C, p.Ile291Thr (in *GSN*) were predicted as probably damaging by the CADD and Polyphen 2 tools. However, variant c.9215G>T, p.Gly3072Val in the *AKAP9* gene was assessed by prediction tools as a neutral influence variant. These variants need to be further evaluated to determine the extent of their impact.

## 4. Discussion

In this study, 40 Vietnamese victims who were diagnosed with SUD at ages < 40 years old were sequenced using Targeted NGS with a panel gene consisting of 167 cardiac disease-associated genes. Fifteen (37.5%) victims were identified with 17 variants in genes associated with cardiomyopathy and channelopathies (Table 2). Molecular forensics studies in young individuals with negative autopsies have identified putative pathogenic variants in genes associated with channelopathies in 11 to 26% of cases [11,42]. In 2014, Bagnall et al. [43] first performed whole exome sequencing (WES) in 28 cases of adolescent SUDs and identified three rare variants associated with LQTS and six variants related to channelopathies and cardiomyopathy. In another study, the authors performed gene panel analysis (including 69, 98, or 101 genes) on 51 SUD cases and WES on another 62 SUD cases, finding variants in 31 cases (27%) [36]. Hata et al. [44] used a panel of 70 genes to evaluate 25 SUD cases and identified five known variants and 10 novel variants predicted to be pathogenic by in silico analysis. The variants included three channelopathy-associated genes (*RYR2*, *CACNA1C*, and *ANK2*), three HCM- or DCM-associated genes (*MYH7*, *LDB3*, and *PRKAG2*), five ACM-related genes (*PKP2*, *JUP*, *DSG2*, *DSP*, and *TMEM43*), and two cardiac transcription factor genes (*TBX5* and *GATA4*). The authors also identified the simultaneous presence of two heterozygous variants in 3 of 25 cases and 2 cases carrying three or more variants [44]. These data support the hypothesis that the “single gene disease” model may not apply to all cases of SUD, which can sometimes occur because of the interaction of multiple variants [45].

Another study performed with a gene panel of 100 genes in 61 SUD cases found that 21 (34%) individuals carried variants that may have a functional effect. Ten (40%) of these variants were in genes associated with cardiomyopathy and fifteen (60%) were in genes associated with cardiac channelopathies [8]. Previous reports suggested that cardiomyopathy often has a variety of manifestations due to incomplete penetrance, so initial phenotypic changes may not be seen at autopsy or may be considered nonspecific or within the normal range. Variants associated with cardiomyopathy and structural alterations of the heart can give rise to arrhythmias—and in some cases, cause disorders through regulation of cardiac channel function [46,47,48,49]. Primary electrical disorders are thought to be caused by variants in genes encoding ion channels and cardiomyopathy [50].

Cerrone and Priori’s study also provided evidence for an association between channelopathies and hereditary cardiomyopathy with arrhythmogenic substrate-related genes thought to be involved in abnormal structures of the myocardium and primary electrical disorders [51]. In these cases, an autopsy showed that the heart had a normal structure and that the arrhythmia arose from abnormalities in the electrical function of the heart [50]. These primary electrical diseases are mainly caused by variants in the genes encoding the heart’s ion channels and receptors. Most variants that lead to ion channel dysfunction alter the electrical activity in the heart and predispose patients to fatal arrhythmias without changing the morphology of the heart tissue. Identifying genetic factors that lead to SCD is important because genetic testing can contribute to diagnosis and risk prediction.

In our study, five cases carried a known pathogenic variant in the following genes: *TNNI3K* (c.2302G>C, p.Glu768Gln, RCV000768402.5/Pathogenic, P3); *HTRA1* (c.496C>T, p.Arg166Cys, RCV002291765.4/Pathogenic, P6); *TNNT2* (c.452G>A, p.Arg151Gln, RCV000796707.6/Pathogenic, P8); *SCN10A* (c.2158G>A, p.Asp720Asn, VCV000532067.8/Pathogenic, P9); and *SCN5A* (c.515A>G, p.His184Arg, VCV000201540.5/Likely pathogenic, P12). Another four patients carried known variants in the following genes: *KCNA5* (c.683C>A, p.Pro228His, VCV002202820.2, P4); *MYBPC3* (c.2275G>A, p.Glu759Lys, VCV000843772.16, P5); *CSRP3* (c.298C>T, p.Arg100Cys, VCV000851709.9, P10); and *TNNI3* (c.292C>G, p.Arg98Gly, VCV001331910.2, P13). Based on the prediction results using prediction tools, three of these variants (*KCNA5*, P4; *MYBPC3*, P5; *TNNI3*, P13) were predicted to be likely to cause disease in the cases. However, the variant c.298C>T, p.Arg100Cys in the *CSRP3* gene was not confirmed as the cause of the disease in case P13 (A in Table 3).

Among four cases that carried novel variants in the following genes: *RYR2* (c.10498G>T, p.Asp3500Tyr, P1), *AKAP9* (c.5187_5188dup, p.Arg1730llefsTer4, P2), *MYH6* (c.1454A>T, p.Lys485Met, P7), and *GSN* (c.872T>C, p.Ile291Thr, P14), three variants (in the *RYR2*, *AKAP9*, and *MYH6* genes) were evaluated as like pathogenic variants. Case P11 carried the following novel variants: c.4840G>A, p.Glu1614Lys (in the *MYLK* gene) and c.9215G>T, p.Gly3072Val (in the *AKAP9* gene); however, variant c.4840G>A, p.Glu1614Lys in the *MYLK* gene was assessed as potentially damaging by CADD, version 1.7 (score: 36.0) and Polyphen version 2 (score: 0.611) software but as neutral by PhD-SNP (score: RI2) software. The variant c.9215G>T, p.Gly3072Val in the *AKAP9* gene was evaluated as a neutral variant by CADD (score: 17.6), FATHMM (score: 3.76), and PhD-SNP (score: RI8) software while being evaluated as potentially harmful by Polyphen 2 (score: 0.973). Likewise, case P15 also carried the following novel variants: c.2535+1G>A (in the *SLC4A3* gene) and c.5025A>T, p.Glu1675Asp (in the *LAMA4* gene). The variant c.2535+1G>A in the *SLC4A3* gene was evaluated as “exon-skipping” by the EX-SKIP software, “donor loss” by NetGene 2 version 2.42 and Spliceailookup software and as a damaging variant by MaxEntScan software (B in Table 3 and Appendix A). The variant c.5025A>T, p.Glu1675Asp in the *LAMA4* gene was assessed as a disease variant by PhD-SNP (score: RI4) and “probably damaging” by CADD (score: 23.6) and Polyphen 2 (score: 0.998), but it was assessed as a “tolerated variant” by FATHMM (score: −1.32) and “benign” by Mutation taster (score: 21) (A in Table 3). Based on the above-predicted results, the c.2535+1G>A variant in the *SLC4A3* gene can be considered the cause of the disease in case P15.

Tester et al. [52] also showed that at least one-third of SUD cases in young people were found to be due to variants in the receptor gene, of which variants in cardiac ryanodine (RYR2) account for nearly 14% of cases. *AKAP9* variants were associated with types of cardiovascular including LQTS type 11 [53,54,55,56], Brugada syndrome [57,58], sudden death of unknown cause [59,60,61,62], severe ventricular arrhythmias [63,64], and cardiomyopathy [65]. Huynh and colleagues confirmed that *AKAP9* variants are associated with fatal arrhythmias and sudden cardiac death [9]. Of interest, some patients may carry the *AKAP9* variant and other genes associated with channelopathies, suggesting the complexity of multifactorial and polygenic inheritance in these patients [54,61]. The variant in the *SLC4A3* anion exchanger gene is considered a cause of SQTS in the patients [66]. In Christiansen’s study, about a quarter of patients with SQTS carried potentially pathogenic variants in *SLC4A3*, representing the most common variants and emphasizing the importance of the inclusion of *SLC4A3* in the genetic screening of patients with SQTS or onset sudden cardiac death [67].

In addition, stroke is the main cause of death in young people. However, the actual genetic cause of stroke is still unknown. Cerebral small vessel disease (CSVD), often causing stroke, is characterized by young adult onset [68] and is considered to be caused by variants in the high temperature-demanding serine peptidase A1 (*HTRA1*) gene [69]. The *HTRA1* gene encodes a serine enzyme that mediates cell signaling and protein degradation and plays an important role in vascular integrity [70]. Autosomal dominant genetic variants on *HTRA1* have been reported in families with CSVD in Italy, Spain, Greece, and China [61,71,72,73]. Heterozygous HTRA1 variants may be responsible for impaired HTRA1 protease activity or form stabilizers leading to the phenotype in patients [74].

In this study, using Targeted NGS sequencing and prediction tools, we identified variations in related genes in 15 SUD cases out of a total of 40 studied cases. Of these 15 cases, the cause of sudden death in 12 cases was identified, and three cases (P10, P11, and P14) required further evaluation to determine the exact cause of sudden death. Forensic molecular autopsy has recently proven valuable in SUD cases and is increasingly used to explain these cases.

There are some limitations in our study. For the first limitation, we used a gene panel with 167 genes related to cardiomyopathy and channelopathies which was reported previously, leading to the possibility of not detecting the variants in other related genes. The second limitation of our study is that we did not conduct studies to evaluate the influence of variants, so we have not yet reached accurate conclusions about the cause of SUD in some cases.

## 5. Conclusions

In our study, 40 SUD cases were investigated with Targeted NGS sequencing, and 17 variants in genes associated with cardiomyopathy and channelopathies in 15 cases were identified. By predicting the pathogenicity of the variants found using prediction software, the cause of death was determined in 12 cases. The obtained results showed that next-generation sequencing is a tool that can be used in molecular forensics to determine the cause of disease in SUD cases.

## Figures and Tables

**Figure 1 diagnostics-14-01876-f001:**
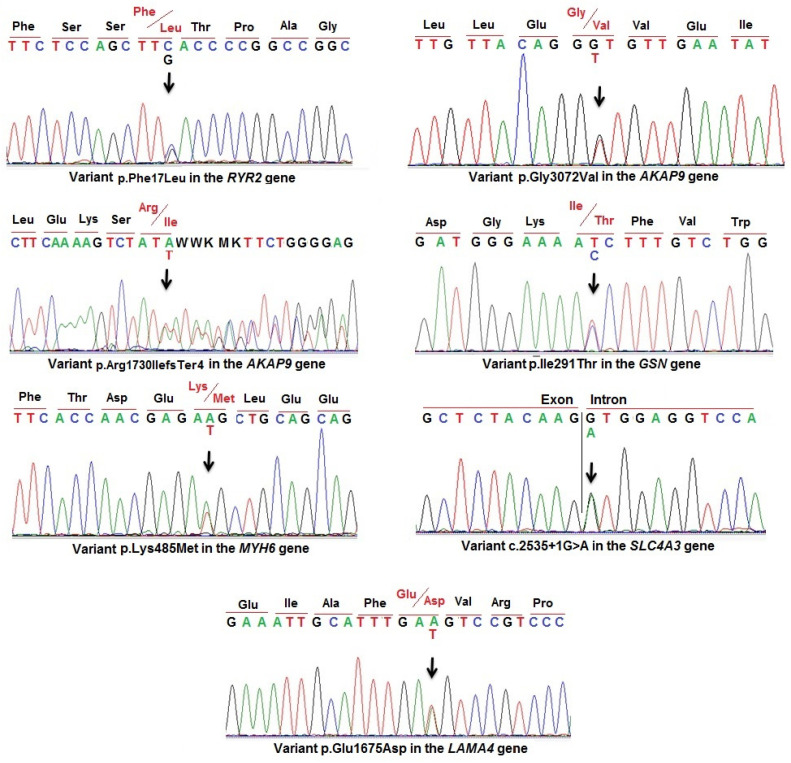
Sanger sequencing results confirming the novel variants in victims. Red letters are the positions of the altered amino acids.

**Figure 2 diagnostics-14-01876-f002:**
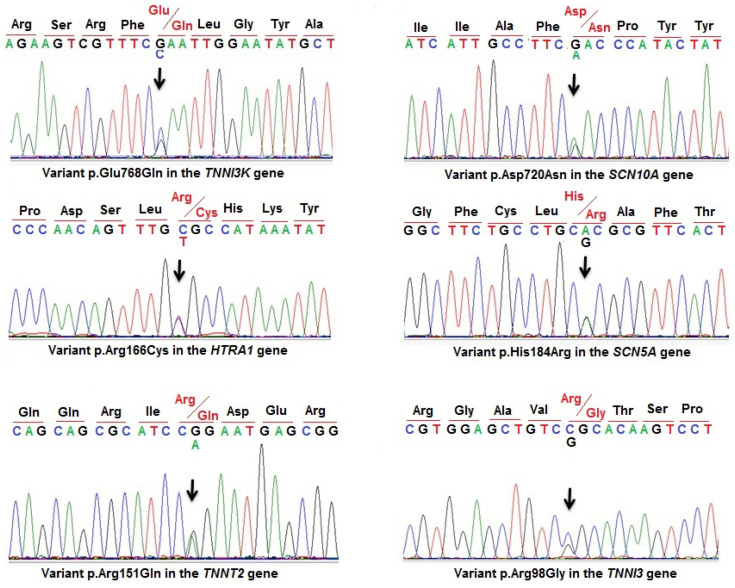
Sanger sequencing results confirming the known variants in victims.

**Table 1 diagnostics-14-01876-t001:** Characteristics of sudden unexplained cardiac death cases.

ID	Sex/Age	Event at SUD	Medical History		Autopsy Findings
			Height (cm)	Health	
P1	F/14	Sleep at night	160	Normal	Negative
P2	M/29	Sleep at night	178	Normal	Negative
P3	M/25	Driving/daytime	158	Normal	Negative
P4	M/19	Sleep at night	175	Normal	Negative
P5	M/28	Sleep at night	174	Normal	Negative
P6	M/40	Eating at night	152	Normal	Negative
P7	M/39	Sleep at daytime	170	Normal	Negative
P8	M/22	Rest at daytime	162	Normal	Negative
P9	M/29	Sleep at night	168	Normal	Negative
P10	F/33	Sleep at night	158	Normal	Negative
P11	F/25	Rest at daytime	156	Normal	Negative
P12	M/35	Sleep at night	170	Normal	Negative
P13	M/39	Sleep at night	165	Normal	Negative
P14	M/24	Working/daytime	180	Normal	Negative
P15	M/23	Sleep at night	168	Normal	Negative
P16	M/27	Working/daytime	160	Normal	Negative
P17	M/30	Working/daytime	172	Normal	Negative
P18	F/1	Rest at daytime	76	Normal	Negative
P19	M/30	Sleep at night	169	Normal	Negative
P20	M/40	Sleep at night	172	Normal	Negative
P21	F/32	Rest at daytime	159	Normal	Negative
P22	M/27	Working/daytime	160	Normal	Negative
P23	M/34	Sleep at night	167	Normal	Negative
P24	M/40	Sleep at night	162	Normal	Negative
P25	M/38	Sleep at night	170	Normal	Negative
P26	M/40	Rest at daytime	162	Normal	Negative
P27	M/35	Sleep at daytime	164	Normal	Negative
P28	F/2	Rest at daytime	80	Normal	Negative
P29	M/33	Rest at daytime	162	Normal	Negative
P30	M/27	Sleep at night	173	Normal	Negative
P31	M/35	Sleep at night	170	Normal	Negative
P32	M/32	Sleep at night	162	Normal	Negative
P33	M/33	Sleep at night	167	Normal	Negative
P34	M/39	Playing a sport	166	Normal	Negative
P35	M/24	Sleep at night	179	Normal	Negative
P36	M/19	Working/daytime	169	Normal	Negative
P37	F/29	Rest at daytime	155	Normal	Negative
P38	F/20	Sleep at night	158	Normal	Negative
P39	M/23	Sleep at night	168	Normal	Negative
P40	F/40	Sleep at night	159	Normal	Negative

**Table 2 diagnostics-14-01876-t002:** Variants found in SUD victims.

ID	Gene	cDNA/Protein	dbSNP/MAF/ClinVar/ExAC	Zygosity
P1 (F/14)	*RYR2* (NM_001035.3)	c.51C>G p.Phe17Leu	novel	het
P2 (M/29)	*AKAP9* (NM_005751.4)	c.5187_5188dup p.Arg1730llefsTer4	novel	het
P3 (M/25)	*TNNI3K* (NM_015978.3)	c.2302G>C p.Glu768Gln	rs202238194/0.00000 RCV000768402.5/pathogenic	het
P4 (M/19)	*KCNA5* (NM_002234.4)	c.683C>A p.Pro228His	VCV002202820.2 uncertain significance	het
P5 (M/28)	*MYBPC3* (NM_000256.3)	c.2275G>A p.Glu759Lys	rs750810342/0.00002487 VCV000843772.16/uncertain significance	het
P6 (M/40)	*HTRA1* (NM_002775.5)	c.496C>T p.Arg166Cys	rs2097494390/VCV001325819.5 RCV002291765.4/pathogenic	het
P7 (M/39)	*MYH6* (NM_002471.4)	c.1454A>T p.Lys485Met	novel	het
P8 (M/22)	*TNNT2* (NM_001276345.2)	c.452G>A p.Arg151Gln	rs730881101/0.00000 RCV000796707.6/pathogenic	het
P9 (M/29)	*SCN10A* (NM_006514.4)	c.2158G>A p.Asp720Asn	rs781354273/0.00006/VCV000532067.8 pathogenic	het
P10 (F/33)	*CSRP3* (NM_003476.5)	c.298C>T p.Arg100Cys	rs201214593/0.00004/VCV000851709.9 uncertain significance	het
P11 (F/25)	*MYLK* (NM_053025.4)	c.4840G>A p.Glu1614Lys	novel	het
	*AKAP9* (NM_005751.4)	c.9215G>T p.Gly3072Val	novel	het
P12 (M/35)	*SCN5A* (NM_00335.5)	c.515A>G p.His184Arg	rs794728898/0.000102/VCV000201540.5 likely pathogenic	het
P13 (M/39)	*TNNI3* (NM_000363.5)	c.292C>G p.Arg98Gly	rs730881068/0.00005/VCV001331910.2 uncertain significance	het
P14 (M/24)	*GSN* (NM_198252.3)	c.872T>C p.Ile291Thr	novel	het
P15 (M/23)	*SLC4A3* (NM_005070.4)	c.2535+1G>A	novel	het
	*LAMA4* (NM_001105206.3)	c.5025A>T p.Glu1675Asp	novel	het

**Table 3 diagnostics-14-01876-t003:** (A) Predictions from in silico software for variants. (B) Predictions from in silico software for the splice variant SLC4A3: c.2535+1G>A.

(A)							
Gene/Variants	ClinVar	CADD Score/Prediction	FATHMM Score/Prediction	Mutation Taster Score/Prediction	PhD-SNP Score/Prediction	Polyphen 2 Score/Prediction	SNP&GO Score/Prediction
*RYR2* c.10498G>T p.Asp3500Tyr	Novel	25.5 Probably damaging	−4.21 Damaging	160 Deleterious	Neutral RI8	1.000 Possibly damaging	Disease RI1
*KCNA5* c.683C>A p.Pro228His	Uncertain significance	24.1 Probably damaging	−0.21 Tolerated	77 Deleterious	Neutral RI5	1.000 Possibly damaging	Disease RI7
*MYBPC3* c.2275G>A p.Glu759Lys	Uncertain significance	26.7 Probably damaging	−0.20 Tolerated	56 Deleterious	Neutral RI5	1.000 Possibly damaging	-
*MYH6* c.1454A>T p.Lys485Met	Novel	28.9 Probably damaging	−2.60 Damaging	-	Disease RI6	1.000 Possibly damaging	Disease RI9
*CSRP3* c.298C>T p.Arg100Cys	Uncertain significance	28.5 Probably damaging	−0.28 Tolerated	89 Benign	-	0.110 Benign	Disease RI2
*MYLK* c.4840G>A p.Glu1614Lys	Novel	36.0 Probably damaging	-	-	Neutral RI2	0.611 Possibly damaging	-
*AKAP9* c.9215G>T p.Gly3072Val	Novel	17.6 Tolerated	3.76 Tolerated	-	Neutral RI8	0.973 Possibly damaging	-
*TNNI3* c.292C>G p.Arg98Gly	Uncertain significance	26.3 Probably damaging	−3.49 Damaging	125 Deleterious	Neutral RI1	1.000 Possibly damaging	Disease RI10
*GSN* c.872T>C p.Ile291Thr	Novel	29.1 Probably damaging	−0.34 Tolerated	Benign	Neutral RI7	1.000 Possibly damaging	-
*LAMA4* c.5025A>T p.Glu1675Asp	Novel	23.6 Probably damaging	−1.32 Tolerated	21 Benign	Disease RI4	0.998 Possibly damaging	Neutral RI7
**(B)**							
**In Silico Prediction Tools**		**Wildtype**	**Mutant**	**Prediction**			
EX-SKIP			−218.867	Exon-skipping			
Fruitfly		NA	NA	-			
MaxEntScan		1.51	−6.67	Damage variant			
NetGene2		0.60	0.00	Donor loss			
Spliceailookup		0.98	0.00	Donor loss			

## Data Availability

Data is contained within the article or Appendix A.

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
