# Peer review of "Identifying the Pathogenic Variants in Heart Genes in Vietnamese Sudden Unexplained Death Victims by Next-Generation Sequencing"

_diagnostics, 2024, doi:10.3390/diagnostics14171876_

Round 1
Reviewer 1 Report
Comments and Suggestions for Authors
Table 1 presents data for 15 cases, although a total of 40 people were included in the study. The description should include the proportion of men and women, the minimum and maximum age, and the median age.
The authors used a panel of 167 genes. This limits the ability to search for causal variants. The number of genes whose changes may be the cause of SUD is constantly increasing, so the best option would be exome sequencing. Unlike genomic sequencing, exome sequencing is not significantly different in cost and technical complexity from a panel of genes. However, it provides an opportunity to re-analyse the data when new information about new genes involved in the development of SUD becomes available. In other words, it should be noted in the discussion that the gene panel can be used, but it is not the best choice.
Author Response
Reviewer #1:
Table 1 presents data for 15 cases, although a total of 40 people were included in the study. The description should include the proportion of men and women, the minimum and maximum age, and the median age.
Thank you so much for your advice. We would like to add information about the 40 victims who were used in the study as suggested by the Reviewer.
The authors used a panel of 167 genes. This limits the ability to search for causal variants. The number of genes whose changes may be the cause of SUD is constantly increasing, so the best option would be exome sequencing. Unlike genomic sequencing, exome sequencing is not significantly different in cost and technical complexity from a panel of genes. However, it provides an opportunity to re-analyse the data when new information about new genes involved in the development of SUD becomes available. In other words, it should be noted in the discussion that the gene panel can be used, but it is not the best choice.
Thank you so much for your advice. We completely agree with the comments of the reviewer. We also recognized the limitations of the gene panel analysis method (although in this study we used a gene panel of 167 genes) and that was mentioned in the limitations of the study. However, due to the limited funding for research, we only used this method in our study. We would like to recommend the use of WES for similar studies as suggested by the Reviewer.
Reviewer 2 Report
Comments and Suggestions for Authors
The authors present an interesting study of genetic testing on 40 cases of sudden cardiac death of under 40-year-old subjects. The introduction of the study should be reduced in its length. In particular, I would suggest moving the list of variants only to the discussion section. Additionally, it would be useful to emphasize in the conclusions that this method, in addition to allowing the identification of the cause of death, enables the genetic screening of individuals potentially at risk. The design of the study is well presented and explained. The results are exhaustively reported, though some parts could be difficult to comprehend. The integration of visual supports could be useful for better clarity and readability. The discussion is extensive and thoroughly integrates the results. This study represents a useful contribution to the field.
Author Response
Reviewer #2:
The authors present an interesting study of genetic testing on 40 cases of sudden cardiac death of under 40-year-old subjects.
The introduction of the study should be reduced in its length. In particular, I would suggest moving the list of variants only to the discussion section.
Thank you so much for your advice. In the introduction we provide an overview of the disease and the genes involved to see the complexity of the underlying causes of sudden death in victims, thereby emphasizing the need to conduct forensic molecular analysis to search for possible causes that can explain these cases. The introduction has been shorted as suggested by the Reviewer.
Additionally, it would be useful to emphasize in the conclusions that this method, in addition to allowing the identification of the cause of death, enables the genetic screening of individuals potentially at risk.
Thank you so much for your advice. The conclusions have been edited as suggested by the Reviewer.
The design of the study is well presented and explained. The results are exhaustively reported, though some parts could be difficult to comprehend. The integration of visual supports could be useful for better clarity and readability. The discussion is extensive and thoroughly integrates the results. This study represents a useful contribution to the field.
Thank you so much for your comments.